# In Vitro and Computational Studies of Perezone and Perezone Angelate as Potential Anti-Glioblastoma Multiforme Agents

**DOI:** 10.3390/molecules27051565

**Published:** 2022-02-26

**Authors:** Maricarmen Hernández-Rodríguez, Pablo I. Mendoza Sánchez, Joel Martínez, Martha E. Macías Pérez, Erika Rosales Cruz, Teresa Żołek, Dorota Maciejewska, René Miranda Ruvalcaba, Elvia Mera Jiménez, María I. Nicolás-Vázquez

**Affiliations:** 1Laboratorio de Cultivo Celular, Escuela Superior de Medicina, IPN, Salvador Díaz Mirón esq. Plan de San Luis s/n, Casco de Santo Tomas, Miguel Hidalgo, Ciudad de Mexico 11340, Mexico; dra.hernandez.ipn@gmail.com; 2Departamento de Ciencias Químicas, Facultad de Estudios Superiores Cuautitlán Campo 1, UNAM. Av. Primero de Mayo S/N, Sta María Guadalupe las Torres, Cuautitlán Izcalli 54740, Mexico; pimendozas1@gmail.com (P.I.M.S.); mirruv@yahoo.com.mx (R.M.R.); 3Facultad de Ciencias Químicas, Universidad Autónoma de San Luis Potosí, San Luis Potosí 78210, Mexico; atlanta126@gmail.com; 4Unidad de Investigación Biomédica de Zacatecas (UIBMZ) del Instituto Mexicano del Seguro Social (IMSS), Alameda Trinidad García de La Cadena 438_2436A436, Zacatecas Centro, Zacatecas 98000, Mexico; marthita_e23@yahoo.com.mx; 5Laboratorio de Hematopatología, Escuela Nacional de Ciencias Biológicas, IPN, Prolongación de Carpio y, Calle Plan de Ayala s/n, Santo Tomás, Miguel Hidalgo, Ciudad de Mexico 11340, Mexico; erika_encb@hotmail.com; 6Department of Organic Chemistry, Faculty of Pharmacy, Medical University of Warsaw, Żwirki i Wigury 61, 02-091 Warszawa, Poland; teresa.zolek@wum.edu.pl (T.Ż.); dorota.maciejewska77@gmail.com (D.M.)

**Keywords:** phyto-compounds, computational studies, drug-likeness, anti-neoplastic activity, glioblastoma multiforme

## Abstract

Glioblastoma multiforme (GBM) represents the most malignant type of astrocytoma, with a life expectancy of two years. It has been shown that Poly (ADP-ribose) polymerase 1 (PARP-1) protein is over-expressed in GBM cells, while its expression in healthy tissue is low. In addition, perezone, a phyto-compound, is a PARP-1 inhibitor with anti-neoplastic activity. As a consequence, in the present study, both in vitro and computational evaluations of perezone and its chemically related compound, perezone angelate, as anti-GBM agents were performed. Hence, the anti-proliferative assay showed that perezone angelate induces higher cytotoxicity in the GBM cell line (U373 IC_50_ = 6.44 μM) than perezone (U373 IC_50_ = 51.20 μM) by induction of apoptosis. In addition, perezone angelate showed low cytotoxic activity in rat glial cells (IC_50_ = 173.66 μM). PARP-1 inhibitory activity (IC_50_ = 5.25 μM) and oxidative stress induction by perezone angelate were corroborated employing in vitro studies. In the other hand, the performed docking studies allowed explaining the PARP-1 inhibitory activity of perezone angelate, and ADMET studies showed its probability to permeate cell membranes and the blood–brain barrier, which is an essential characteristic of drugs to treat neurological diseases. Finally, it is essential to highlight that the results confirm perezone angelate as a potential anti-GBM agent.

## 1. Introduction

Glioblastoma multiforme (GBM) represents the most malignant type of astrocytoma, which is distinguished by its high ability to produce metastasis. Patients with GBM have the worst prognosis [1,2,3,4]. Nowadays, GBM treatment includes surgery, radiotherapy, and chemotherapy, which have shown minimal improvements [5]. Moreover, local GBM infiltration into normal tissue results in tumor recurrence, culminating with the patient’s death [6]. Hence, chemotherapy represents a valuable tool to treat GBM. Nevertheless, its efficacy is diminished by the blood–brain barrier (BBB) that restring the diffusion of therapeutic compounds to the brain [7]. The principal treatments employed only increased the life expectancy of GBM patients to fifteen months [8]. Consequently, the design of novel directed therapies for GBM results are of great interest.

Poly ADP-ribose polymerase 1 (PARP-1) protein represents a promising target for developing of anti-neoplastic compounds. PARP-1 is best known for its role in deoxyribonucleic acid (DNA) single-strand breaks repair; therefore, PARP-1 inhibition avoids DNA repair and the induction of death of cancer cells [9].

Recently, immunohistochemical studies showed that PARP-1 is overexpressed in GBM cells, being practically undetectable in normal brain tissue [10]. This fact has allowed the search for novel diagnostic tools for GBM by synthesizing a small molecule based on radio iodinated PARP-1 targeted tracers [11]. Additionally, higher PARP-1 levels have exhibited an inverse correlation with patient survival [12]. This fact emphasizes the importance of PARP-1 inhibitors as a potential drug for the treatment of GBM. In this sense, perezone, a phyto-compound extracted from roots of the plant of the genus *Acourtia*, has exhibited anti-neoplastic activity by PARP-1 inhibition and reactive oxygen species release [13]. For this reason, the present research work aimed to perform in vitro and computational evaluation of perezone and its chemically related compound, perezone angelate (Figure 1), as anti-GBM agents.

## 2. Results

### 2.1. Extraction of Perezone and Perezone Angelate

The extraction of perezone and perezone angelate from roots of Acourtia cordata was performed as previously described [13]. Physical and spectroscopic data were correlated with the literature [14].

Perezone: Yellow crystalline solid, R_f_ = 0.42 (*n*-hexane/AcOEt 9:1), mp = 104 ± 1 °C, ^1^H NMR (300 MHz, CDCl_3_) δ (ppm): 1.20 (d, *J* = 7.1 Hz, 3H), 1.53 (s, 3H), 1.58 (m, 1H), 1.64 (s, 3H), 1.80 (m, 1H), 1.86 (dt, *J* = 21.1, 7.0 Hz, 1H), 1.92 (m, 1H), 2.06 (d, *J* = 1.6 Hz, 3H), 3.05 (m, 1H), 5.07 (dd, *J* = 10.0, 4.1 Hz, 1H), 6.48 (d, *J* = 1.6 Hz, 1H), 6.99 (s, 1H); ^13^C NMR (75 MHz, CDCl_3_) δ (ppm): 14.71, 17.63, 18.24, 25.70, 26.69, 29.33, 34.11, 124.48, 124.59, 131.45, 135.88, 140.55, 150.98, 184.34, 187.39; ESI-HRMS (19 eV), exact mass for C_15_H_21_O_3_, [M + H]^+^ 249.14907 Da, corresponding accurate value: 249.14966 Da, error: + 0.59, insaturations: 5.5.

Perezone angelate: Yellow oil, R_f_ = 0.49 (n-hexane/AcOEt 9:1), ^1^H NMR (300 MHz, CDCl_3_) δ (ppm): 1.08 (d, *J* = 7.2 Hz, 3H), 1.53 (s, 3H), 1.60 (m, 1H), 1.63 (s, 3H), 1.82 (m, 1H), 1.90 (dt, *J* = 21.3, 7.0 Hz, 1H), 1.92 (m, 1H), 2.05 (quintet, *J* = 1.5, 3H), 2.06 (d, *J* = 1.6 Hz, 3H), 2.07 (dd, *J* = 7 Hz, 1.5 Hz, 3H), 2.99 (m, 1H), 5.02 (dd, *J* = 10.0, 4.1 Hz, 1H), 6.33 (q, *J* = 7 Hz, 1H), 6.48 (d, *J* = 1.6 Hz, 1H); ^13^C NMR (75 MHz, CDCl_3_) δ (ppm): 15.27, 15.98, 17.67, 18.63, 20.50, 22.41, 25.69, 26.55, 34.68, two overlapping signals at 124.03, 126.08, 131.97, 134.11, 139.92, 142.43, 143.76, 170.16, 180.73, 186.89; ESI-HRMS (19 eV), exact mass for C_20_H_27_O_4_, [M + H]^+^ 331.1904 Da, corresponding accurate value: 331.1945 Da, error: 9.3796, unsaturations: 7.5.

### 2.2. Perezone Angelate Showed Greater Cytotoxic Activity in U373 Cells than Normal Rat Glial Cells

Cytotoxic evaluation of perezone and perezone angelate was performed in U373 cells, considering its pro-apoptotic effect, as previously reported [13], employing suberoylanilide hydroxamic acid as a positive control. Figure 2a shows the cytotoxic activity of suberoylanilide hydroxamic acid, with an 85.8% viability at 10 μM, similar to data previously reported [15]. According to the results, it is possible to observe that perezone and perezone angelate showed cytotoxic activity dependent on concentration (Figure 2a,b). The cytotoxic activity of perezone angelate (IC_50_ of 6.44 ± 1.6 μM) was higher (Figure 2a) in comparison to perezone (IC_50_ of 51.20 ± 0.3 μM) in the U373 cell line. For mixed glial cell culture, perezone showed an IC_50_ of 59.85 ± 0.3 μM (Figure 2b), which was very similar to that found in the U737, while perezone angelate showed an IC_50_ of 173.66 ± 1.6 μM (Figure 2b). However, although perezone angelate showed cytotoxicity in mixed glial cell culture, the IC_50_ of this compound was twenty-six times lower in U373 cells, thus demonstrating its anti-neoplastic effect.

### 2.3. Perezone Angelate Induces Apoptosis in U373 Cells

According to the obtained results, it is possible to observe that perezone and perezone angelate increase cells marked with Annexin V-FITC+/7AAD, establishing that cells were under apoptosis (Figure 3a). Furthermore, the percentage of apoptotic cells was statistically significant higher for perezone angelate than for perezone (Figure 3b), and apoptosis induction is preferred over necrosis by anti-neoplastic agents [16]. The achieved results emphasize the importance of perezone angelate.

### 2.4. Perezone and Perezone Angelate Inhibit U373 Cell Migration

Preventing the invasiveness and metastasis in cancer is a challenge that requires urgent solutions, especially in the case of GBM [17]. For this reason, the evaluation of migration of U373 under treatment with perezone angelate and perezone at low concentrations (6.25 μM) was performed. As can be seen in Figure 4, U373 cells in control groups reach confluence and cover the wound space; in contrast, fewer U373 cells migrated in the groups treated with perezone and perezone angelate (Figure 4a), and consequently, the percentage of area covered by migration of the U373 cells treated with studied compounds was significantly lower to the control. However, there were no differences in the percentage of the covered area by migrating cells between both compounds (Figure 4b).

### 2.5. Perezone Angelate Is a PARP-1 Inhibitor

As it can be seen in Figure 5, the potent inhibition of PARP-1 by olaparib was confirmed with an IC_50_ of 5.5 nM (± 0.3 μM), which is in accordance with the literature [18]. Importantly, PARP-1 inhibitory activity by perezone angelate was confirmed, with an IC_50_ of 5.25 μM (± 1.2 μM), in this way, showing a higher potency than perezone [13].

### 2.6. Perezone and Perezone Angelate Increases Reactive Oxygen Species (ROS) Production in U373 Cells

ROS production was measured by the fluorescent DCFH-DA technique. As shown in Figure 6, the treatment of U373 cells with perezone and perezone angelate induces ROS production in a dose-dependent manner with both compounds, compared to the control group. The induction of a more oxidized status was similar for both compounds did not show statistically significant differences.

### 2.7. Perezone Angelate Interacts with the PARP-1 Active Site Revealed by Docking Studies

Traditionally, the design of novel PARP-1 inhibitors is aimed to target several of 50 amino acid residues located at the PARP-1 active site [19]. Since it was reported that PARP-1 presents high stability, the conformer obtained from PDB was employed to perform docking studies [20]. Results displayed that perezone angelate interacts with the catalytic site of PARP-1, as can be seen in Figure 7. As a measure of affinity to PARP-1, ΔG values obtained by each compound are shown in Figure 7. As expected, perezone angelate showed the highest affinity (−9.39 kcal/mol).

The binding modes of perezone and perezone angelate with the catalytic site of PARP-1 are shown in Figure 7. The presence of an angelic substituent in perezone angelate results in an increase in affinity to PARP-1 in comparison with perezone (ΔG = −9.39 and −7.24 kcal/mol, respectively); the energy difference shown between both compounds is 2.15 kcal/mol.

Perezone establishes hydrogen bonds with the lateral chain of Tyr235, Glu327, and Tyr246 and hydrophobic interactions with Ala237 (Figure 7). In comparison, the presence of the angelic substituent in perezone angelate allows to change the binding mode and consequently create non-bonding interactions with Tyr228, Tyr235, Ile234, Hys201, Leu216, Arg217, and Asp105.

### 2.8. Perezone Angelate Showed a High Probability to Permeate Cells and Cross the BBB According to ADMET Predictions

In silico ADMET study was performed for perezone and perezone angelate. The values obtained for Lipinski’s rule of five, topological polar surface area (TPSA), and the solubility (S_w_) are summarized in Table 1. The molecular weights (MWt) are defined for orally available compounds. The calculated logarithm of the 1-octanol-water partition coefficient (log P) values for perezone (3.336) and perezone angelate (4.478) indicate that these can be absorbed (log P < 5). The distribution coefficient (Log D) values, which give an estimate for ionizable forms of a drug, were calculated at a pH of 7.4. The log D values for perezone angelate exceeded the traditional cutoff value of 4.0, indicating a higher probability of binding to plasma proteins [21]. The number of atoms engaged in the intermolecular hydrogen bonding and the number of rotatable bonds are within the usually acceptable ranges. TPSA is a good descriptor of absorption, including intestinal absorption, bioavailability, and BBB penetration [22]. TPSA values for studied compounds were found below 90 Å^2^, thus predicting favorable conditions for the penetration the BBB, which is essential for the interaction of the ligand with brain targets. The water solubility (S_w_) values for studied compounds show acceptable solubility, and we can suppose that the analyzed compounds are potentially promising agents for advanced biological screening.

The predictions of parameters joined with permeability through biological barriers are presented in Table 2. The distribution (V_d_) values of all tested compounds are below 2.7 L/kg, which means that they are predicted to be confined to the blood plasma. Effective permeability (P_eff_), which reflects the passive transport velocity across the epithelial barrier in the human jejunum, shows high values for jejunal permeability (4.547–9.250 × 10^−4^ cm/s). The evaluation of membrane permeability by Madin–Darby Canine Kidney (MDCK) cells shows the values in the range 399–1106.031 × 10^−7^ cm/s indicating the good apparent membrane permeability. The efficiency of a drug may be affected by the degree to which it binds to the proteins within blood plasma. The human serum albumin binding is a good parameter for evaluating drug availability, and consequently, the efficacy of a drug can be affected by the degree of plasma protein binding. Compared with the recommended values of the unbound drug to proteins within blood plasma (%Unbnd) > 10%, the results showed a higher percentage of non-bound parts for perezone angelate than for perezone itself, indicating that they can circulate more freely within the bloodstream and hence can have access to the target site.

The BBB limits the delivery of chemotherapeutic agents to the brain, and the evaluation of BBB penetration is crucial for analyzed substances. Two parameters were calculated: BBB filter and logBB. Two compounds are characterized by a high probability of cross BBB (Table 2). The highest concentration in the brain is predicted for compound perezone angelate, which is very promising. The affinity to P-gp, an efflux pump, can be evaluated as a biological barrier for toxins, xenobiotics, and potential chemotherapeutic agents [24], because it can expel the molecules from the cells out. As it can be seen in Table 2, tested phyto-compounds are not substrates to P-gp, and perezone angelate can even be an inhibitor.

## 3. Discussion

GBM is one of the most lethal and difficult to treat cancers. It has a poor prognosis with maximum 5-year survival is 7.2% of patients. Due to its highly infiltrative nature, one of its characteristics is the invasiveness, and despite maximum resection, GBM tumors recur after treatment [25].

Many factors prevent the development of novel compounds to treat GBM such as the BBB, the intrinsic resistance of GBM cells to the induction of cell death, and its complex pathogenesis [17]. The current therapeutic schedule is aggressive, including surgical resection, temozolomide (TMZ), and concurrent adjuvant radiation therapy, and yet, this strategy only delays tumor progression. Furthermore, it causes significant adverse reactions, reducing the patient quality of life [26]. Thus, low-toxicity, effective drugs/protocols are urgently needed.

Immunohistochemical studies showed that PARP-1 protein is over-expressed in GBM cells, being practically undetectable in normal brain tissue [10]. Additionally, it was demonstrated that PARP-1 is overactive in GBM [27] due to high levels of fragmented DNA [28]. Indeed, higher PARP-1 levels have been exhibited an inverse correlation with patient survival [12]. This observation highlights the importance of developing PARP-1 inhibitors for the treatment of GBM [29].

Natural plant products represent a valuable source of anti-neoplastic compounds. Of all the small compounds employed as anti-neoplastic drugs in the last decade, 49% were phyto-compounds or derivatives. Plant-derived compounds such as alkaloids, taxanes, epipodophyllotoxins, and camptothecins are still the major compounds employed to treat different types of cancers [30].

The cytotoxic activity of perezone angelate was demonstrated, and interestingly, it showed its preference to cancer cells (26 fold), which represents the desired characteristic to novel anti-neoplastic compounds [31]. In addition, U373 cells treated with perezone angelate displayed an increase in Annexin-V(+)/7-AAD(−) dyed, which evidenced that apoptosis explains its cytotoxicity and highlighting its importance as an anti-neoplastic compound. Although, in the past, it has not been believed that GBM does not develop metastasis to extracranial organs due to the presence of the BBB, recently, GBM metastasis outside the central nervous system was demonstrated [32]. In this sense, it is essential to emphasize the migration inhibition exhibited by perezone angelate in U373 cells.

As reported previously, perezone exerts pro-apoptotic effects by at least two mechanisms of action: the induction of oxidative stress and PARP-1 inhibition [13]. Figure 8 shows the proposed effects of perezone angelate to explain its pro-apoptotic effects in GBM cells. Perezone and perezone angelate belong to quinones; due to this chemical characteristic, ROS production could be related to an increase in the activity of nicotinamide adenine dinucleotide phosphate (NADPH) oxidase (Nox4), as it has been demonstrated by other quinone derivatives [33]. Furthermore, NAD(P)H:quinone oxidoreductase (NQO1) has been related to Nox4 stimulation [34]. ROS production allows explaining in part the cytotoxic activity of perezone and perezone angelate due to cancer cells being exposed to a relatively high level of ROS compared to that for normal cells, which is primarily due to their active metabolism driven by oncogenic signals [35]. The pharmacological elevation of intracellular ROS represents an effective strategy to target cancer cells selectively. An exogenous source of ROS insult that is within a tolerable level to normal cells could exceed the threshold that cancer cells can endure, leading to the selective eradication of cancer cells [36]. Consequently, excessive ROS levels irreversibly could damage DNA and lipids and ultimately cause the apoptosis of cancer cells [37,38].

In addition, it has been shown that perezone produced a reduction in the mitochondria membrane potential in a dose-dependent manner that correlates with its cytotoxic activity [39]. In this sense, perezone and probably its derivatives could alter the mitochondrial electron transport, inducing the apoptosis intrinsic pathway.

Although ROS production is very similar by perezone and perezone angelate, the highest inhibition of PARP-1 by perezone angelate explains its high cytotoxic activity. Particularly, cancer cells have defects in DNA repair pathways; then, tumor cells are susceptible to PARP-1 inhibition [40]. Finally, PARP-1 inhibition could influence cell migration, as previously reported [41].

Docking results allows explaining the affinity of perezone angelate to PARP-1 by forming a set of non-bonding interactions (Tyr228, Tyr235, Hys201, Ile234, Leu216, Arg217, and Asp105); in this case, the tyrosine residues can display significant π–π interactions [20] between the aromatic ring of amino acids and the double bond of the side chain. In addition, the acidic protons (OH) of Ile and Arg amino acids contribute to a possible hydrogen bond with the carbonyl groups of the quinone ring. Furthermore, the imidazole moiety of histidine and the carbonyl of aspartic acid can show other meaningful π–π interactions with the double bond of the quinone ring with the double bond of the angelic moiety, respectively. Consequently, all these interactions are more significant than only the one possible π–π interaction between the aromatic ring of Tyr246 amino acid and the double bond of the side chain of perezone, which plays an important role [20]. It is important to emphasize that the presence of an angelic moiety allows perezone angelate to fit deeply in the PARP-1 catalytic site, highlighting that the position of the side chain of perezone angelate is placed in the opposite position in comparison to the side chain of perezone, probably promoting the higher non-interactions above, explaining its improved affinity for PARP-1 in comparison with perezone. The affinity of perezone angelate, exhibited by tight close binding, has been shown for other protein–ligand complexes [42].

Interesting ADMET studies revealed that perezone angelate showed a reasonable probability of being well absorbed, a high probability of permeating cells, and the BBB. This fact acquires importance because the BBB stops 95% of molecules from drug development of neurological disorders [43]. Additionally, the determination of affinity to P-gp is a parameter that needs to be considered during the development of drugs to treat brain diseases. P-gp is an efflux pump that can transport molecules from the cells out. From the acquired computational results, it can be highlight that perezone angelate is not a substrate to P-gp, and it can even be an inhibitor, thus pointing to the importance of this compound as an anti-GBM agent.

## 4. Materials and Methods

### 4.1. General

The perezone and perezone angelate extraction was performed as previously described [13]. Melting points (°C) were determined in a Fisher Jones apparatus Scientific Serial No. 810N0220 (Cole-Parmer, IL, USA). The mass spectrums (ESI 19 eV) were obtained in a microOTOF-Q II mass spectrometer (Agilent Technologies, CA, USA). ^1^H and ^13^C spectra were recorded on a Varian Mercury 300 (Varia, CA, USA)(^1^H 300.08 MHz, ^13^C 75.46 MHz) in CDCl_3_ at room temperature with TMS as the internal reference, chemical shifts (*δ*) are expressed in ppm and the coupling constants (*J*) are expressed in Hz. TLC was performed on precoated silica gel Merck Kieselgel 60 F_254_ plates (Merck KGaA, Darmstadt, Germany) and detected with UV light. All reagents and solvents were analytical grade and were employed without previous purification; they were purchased from Sigma Aldrich Co (Saint Louis, MO, USA).

### 4.2. Anti-Proliferative Evaluation

U373 cells were acquired from ATCC^®^; culture medium, antibiotics, trypsin, and Fetal Bovine Serum (FBS) were acquired from Gibco^®^; and Thiazolyl Blue Tetrazolium Bromide reagent (MTT) was acquired from Sigma Aldrich^®^. The primary mixed glial cell culture was obtained according to procedures reported previously [44]. U373 and microglial cells were cultured in 75 cm^2^ sterile flask with DMEM 12 supplemented with 10% FBS. Cells were maintained in a CO_2_ incubator at 5% at 37 °C for all the stages of experiments. Cytotoxic assays of studied compounds and suberoylanilide hydroxamic acid (10 μM) as a positive control [15] in U373 and mixed glial cells were performed by MTT assays as it was widely reported [45]. U373 and mixed glial cells were removed from the tissue culture flask by adding 3 mL of 0.05% trypsin–EDTA (GIBCO Laboratories) and diluted with fresh media. A cell suspension (100 μL containing 5 × 10^4^ cells) was added to the wells of a 96-well culture plate and incubated for 24 h at 37◦ C in a 5% CO_2_ incubator. Then, 100 μL of solutions with increasing concentrations of perezone and perezone angelate diluted in fresh medium containing ethanol 2% were added to the respective wells, obtaining a final concentrations of 6.25, 12.5, 25, 50, 100, 200, and, 400 μM (*n* = 8) and incubated for an additional 48 h (final concentration of ethanol 1%). At this moment, 20 μL of the MTT reagent (5 mg/mL) was added to each well, and the culture plate was incubated for an additional 3.5 h. At the end of the incubation, the medium culture was removed carefully, and the formazan produced by viable cells was diluted by adding 50 μL of DMSO to each well. The absorbance was measured at 540 nm using a UV/Vis spectrophotometer E max Precision microplate reader (Molecular Devices, CA, USA). Absorbances from cells without treatment were considered as 100% of viability. The concentration–response graphic was obtained for evaluated compounds, and the inhibitory concentration 50 (IC_50_) was calculated by linear regression analysis.

### 4.3. Apoptosis Determination

The apoptotic cell ratio was measured by Annexin V/7AAD staining followed by flow cytometry analysis. Simultaneous staining of cells with Annexin V/7-AAD allows the discrimination of intact cells (Annexin V-/7-AAD-), early apoptotic (Annexin V+/7-AAD —), and late apoptotic or necrotic cells (Annexin V+/7-AAD +) [46]. Briefly, U373 cells (1 mL containing 10^6^ per well) were seeded into 24-well microplates and incubated by 24 h. Then, 1 mL of solutions with increasing concentrations of perezone and perezone angelate diluted in fresh medium containing 2% ethanol were added to the respective wells, obtaining a final concentration of 6.25 and 50 μM (*n* = 3) and incubated for an additional 48 h (final concentration of ethanol 1%), employing cells without treatment as control. At the end of the incubation, cells were trypsinized as previously described and harvested in cytometry tubes. Then, the cells were washed with PBS, suspended in 200 μL of binding buffer, which contains 2.5 μL of Annexin V (AdipoGen International, USA, Catalog Number AG-40B) and 1 μL 7AAD (MbI International Corporation, USA, Catalog Number FP00020050), and gently homogenized and incubated for 15 min at room temperature (25 °C) in the dark. Labeled cells were acquired at 10,000 events in a FACSCalibur flow cytometer (Becton Dickinson, CA, USA), and data were processed by Cell Quest Pro software (Becton Dickinson, CA, USA).

### 4.4. Scratch Wound Assay

Migration of U373 in the presence and absence of perezone and perezone angelate was performed according to the methodology previously described [47]. For this purpose, microglia and U373 of GBM cells were cultured in 24-well microplates (10^5^ per well) until confluence was reached, and a thin “wound” was introduced by scratching with a sterile pipette tip [48]. Then, the cells were washed with PBS and cultured with DMEM with FBS 10% in the absence (control) and presence of target compounds at 6.25 μM (*n* = 3). After 48 h of incubation, pictures were taken by a phase-contrast inverted microscope Olympus IX51.

### 4.5. PARP-1 Inhibition Activity

To corroborate PARP-1 inhibition by perezone angelate, PARP-1 activity was determined in its absence and presence, employing a colorimetric assay kit (BPS Bioscience, USA, Catalog Number 80580), according to manufacturer’s instructions, employing olaparib (BPS Bioscience, CA, USA, Catalog Number 27003) as a reference compound at crescent concentrations (0.00005, 0.0001, 0.0005, 0.001, 0.005, and 0.01 μM, *n* = 3). Perezone angelate was employed at final concentrations of 0.5, 1, 5, 10, 50, and 100 μM (n = 3). Absorbances were obtained at 450 nm using a UV/Vis spectrophotometer E max Precision microplate reader (Molecular Devices, CA, USA).

### 4.6. ROS Production

The increase in ROS production induced in U373 cells by treatment with perezone and perezone angelate was determined employing 2′,7′-dichlorofluorescein diacetate (DCFH-DA, Sigma-Aldrich, MO, USA, Catalog Number D6883) [47]. Briefly, U373 cells were plated in a 24-well tissue culture plate at a density of 10^5^ cells per well. After, the medium was replaced with DMEM containing crescent concentrations of perezone and perezone angelate (12.5, 50, 100, 200, 400, and 800 μM, *n* = 3) and incubated at 37 ˚C in a humidified atmosphere of 5% CO_2_ for 24 h. Following treatment, the cells were harvested and gently washed with PBS. The cells were stained with DCFH-DA (10 μM) and incubated for 30 min at room temperature in the dark to later be analyzed at 530 nm on BD FACSAria 1 flow cytometer. A minimum of 10,000 cells per sample was acquired and analyzed in a FACSCalibur flow cytometer (Becton Dickinson, CA, USA), and the data were processed by Cell Quest Pro software (Becton Dickinson, CA, USA).

### 4.7. Statistical Analysis

The obtained results are presented utilizing mean and standard errors (SE) and were analyzed by one-way ANOVA and Tukey’s post hoc test, stating a statistically significant difference when *p* < 0.05.

### 4.8. Docking Studies

In order to understand the binding mode of perezone angelate, docking studies were achieved, employing perezone as a control. The 2-dimensional structures of the molecules were drawn employing ChemBioDraw Ultra 12.0 (Version 12.0, PerkinElmer, TX, USA) and HyperChem (Hypercube, Inc., FL, USA)was used to pre-optimize their geometry. Subsequently, the complete optimization of the 3-dimensional structures was achieved using the Gaussian 09 program (Gaussian, Inc., PA, USA) [49] at the theory B3LYP/6-311++G(d,p) level [50,51,52,53,54,55]. The PARP-1 catalytic domain was retrieved from the Protein Data Bank (PDB ID: 1UK0), maintaining only one monomer. Since it was reported that PARP-1 presents high stability, the structure obtained from PDB was employed to perform docking studies [20]. Docking studies were performed employing AutoDock 4.2 software due to its high correlation with experimental data. A directed docking was performed employing a rectangular lattice (70 × 70 × 70 Å) with points separated by 0.375 Å that was centered on the active site of PARP-1. All docking simulations were conducted following parameters previously described [56]. The binding poses with the lowest free energy binding (ΔG) were analyzed employing AutoDock tools, and the images were created using PyMol.

### 4.9. ADMET Predictions Details

The molecular structures of perezone and perezone angelate were used as input to the mathematical models implemented in the ADMET Predictor^TM^ version 9 program to generate estimates for each of the drug-likeness properties. In a first instance, the Lipinski rule of five as the filter of toxicity was checked; then, various physicochemical/pharmacokinetic properties such as the topological polar surface area (TPSA), the volume of distribution (V_d_), the solubility (S_w_), the effective permeability (P_eff_), Madin–Darby Canine Kidney cells apparent permeability (MDCK), the percentage of the unbound drug to proteins within blood plasma (%Unbnd), the blood-to-plasma concentration ratio (RBP), the blood–brain barrier (BBB filter), the blood–brain barrier partition coefficient logC_brain_/C_blood_ (logBB), and the P-glycoprotein (P-gp) affinity were estimated for all compounds at pH 7.4.

## 5. Conclusions

Cell proliferation demonstrated that treatment with perezone angelate significantly induces apoptosis in U373 cells, with low cytotoxicity in rat glial cells, being apoptosis the type of cell death. The inhibition of PARP-1 and the induction of an oxidative stress state in U373 cells by perezone angelate were verified.

The performed docking studies showed that perezone angelate established numerous interactions with the catalytic domain of PARP-1, being promoted by the orientation of angelic moiety and the side chain. ADMET studies revealed that perezone angelate exhibited a reasonable probability of being well absorbed and a high probability to permeate cells and the BBB.

Results obtained reveal the anti-GBM activity of perezone angelate, highlighting the importance of the consequent experimental evaluation of this compound by pharmacokinetics assays to ensure that perezone angelate crosses the BBB, pharmacological evaluation, in an in vivo model of GBM, in addition to toxicological evaluation to determinate its adverse side effects.

## Figures and Tables

**Figure 1 molecules-27-01565-f001:**
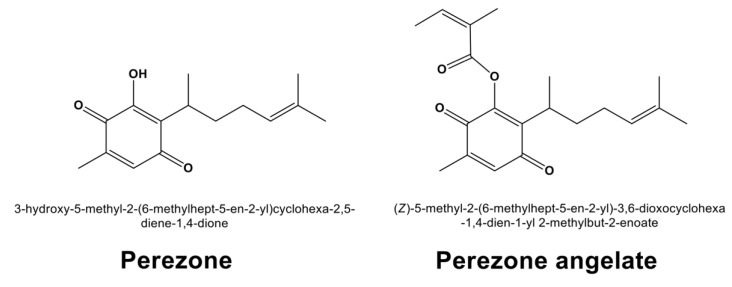
Chemical structures of perezone and perezone angelate.

**Figure 2 molecules-27-01565-f002:**
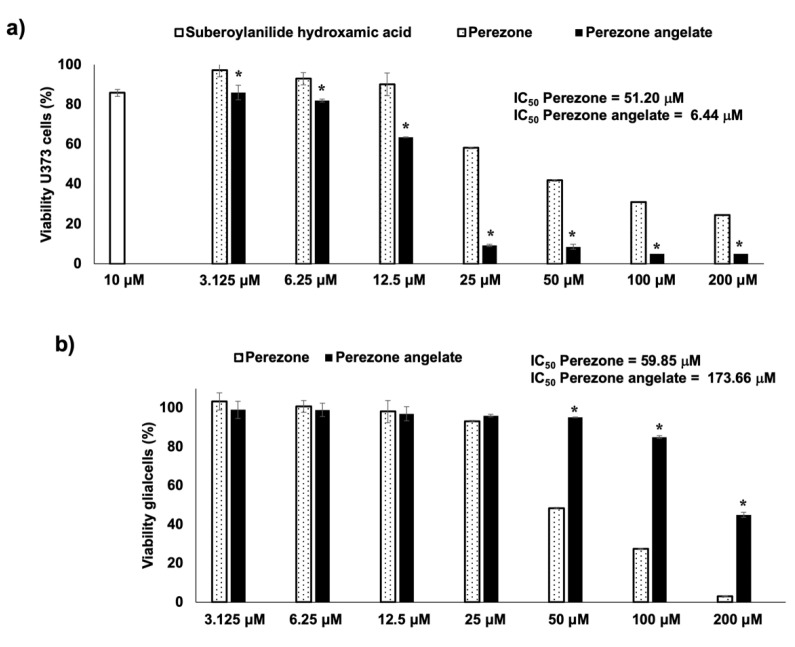
Graph bar of cytotoxic activity of perezone and perezone angelate in the GBM U373 cell line (**a**) and mixed glial cell culture (**b**). U373 cells (5 × 10^4^ cells/well) and mixed glial cells (5 × 10^4^ cells/well) were incubated without (control) and with crescent concentrations of the studied compounds during 48 h (*n* = 8), employing suberoylanilide hydroxamic acid (10 μM) as a positive control. At the end of the incubation, an MTT assay was performed to determine the viability of cells under treatment. Each bar in the graph indicates percentage means and standard error of the mean, statistical differences were calculated by one-way ANOVA * *p* < 0.05, (between perezone and perezone angelate at each concentration of tested compounds). Perezone angelate showed higher cytotoxic activity in U373 cells than perezone and lower cytotoxic activity in mixed glial cells than perezone.

**Figure 3 molecules-27-01565-f003:**
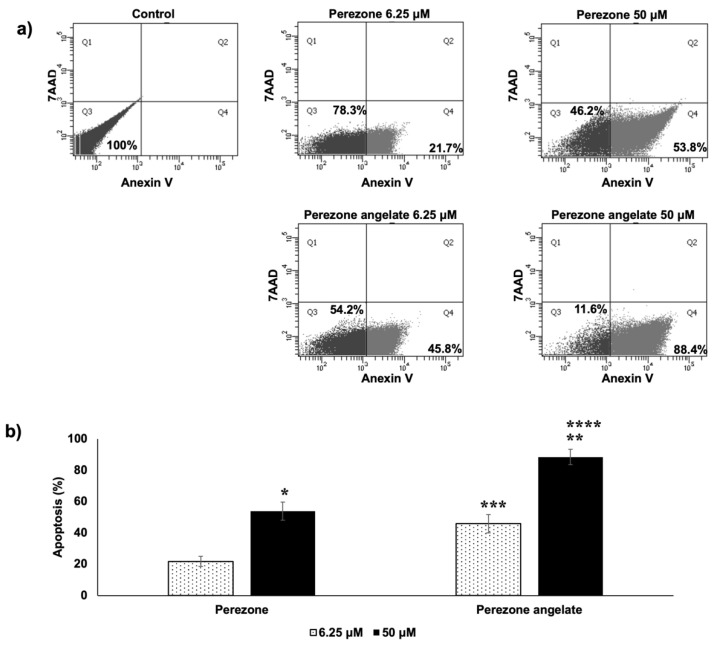
U373 cells were marked with Annexin V-FITC+/7AAD- (apoptotic) under perezone and perezone angelate treatment. Cells were incubated in the absence and presence of studied compounds for 48 h at 6.25 and 50 μM (*n* = 3). After Annexin-V/7-AAD staining, cells were analyzed by flow cytometry. (**a**) Representative histograms and the percentage of apoptotic U373 cells; (**b**) In the graph bar, each one indicates the percentage of mean apoptotic cells ± SEM, statistical differences were calculated by one-way ANOVA followed by a Tukey post hoc test, * *p* < 0.05 between perezone 6.25 μM and perezone 50 μM, ** *p* < 0.05 between perezone angelate 6.25 μM and perezone angelate 50 μM, *** *p* < 0.05 between perezone 6.25 μM and perezone angelate 6.25 μM, **** *p* < 0.05 between perezone 50 μM and perezone angelate 50 μM. The induction of apoptosis was higher to perezone angelate in comparison to perezone.

**Figure 4 molecules-27-01565-f004:**
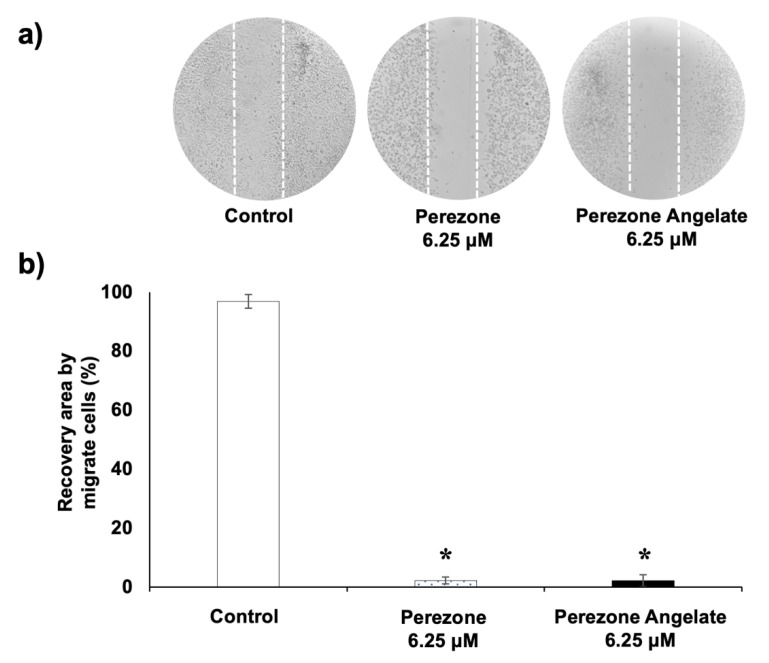
U373 cell migration in vitro is inhibited by perezone and perezone angelate. Wounds with a sterile tip were completed after cells reached confluence. Wound closure was quantified in the absence and presence of studied compounds after 48 h of treatment (*n* = 3): (**a**) Representative image (40×) of U373 cells under treatment; (**b**) the bars express the average percentage of the covered area and the vertical lines express the ± SEM. Statistical analysis was performed by one-way ANOVA and Tukey’s post hoc test * *p* < 0.05 (between control and treatments).

**Figure 5 molecules-27-01565-f005:**
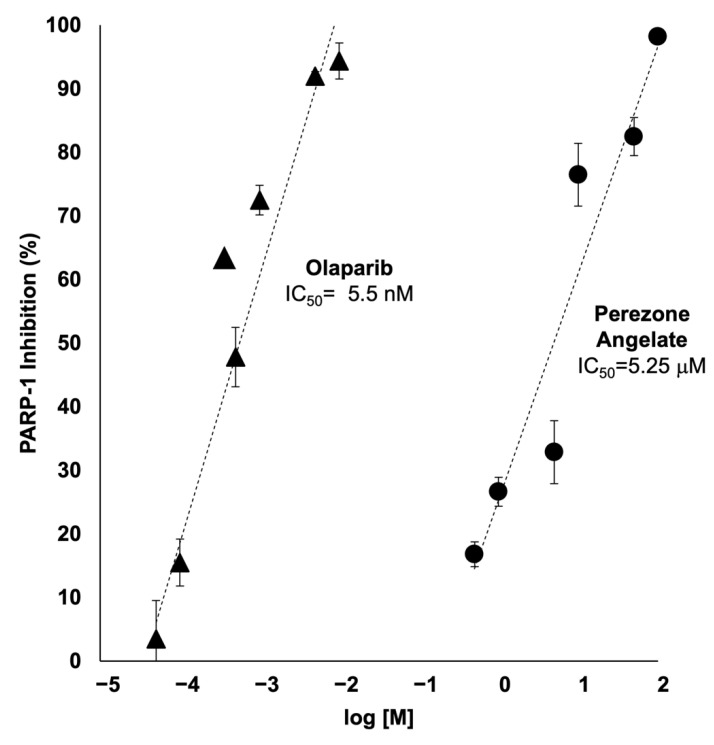
PARP-1 inhibition activity of perezone angelate (circles) and olaparib (triangles). According to the manufacturer’s instructions (PARP-1 colorimetric assay kit BPS Bioscience, USA, Catalog Number 80580). Values on the graph represent mean values, and vertical bars denote standard error (*n* = 3). The half maximal inhibitory concentration (IC_50_) was estimated for each compound through linear regression analysis.

**Figure 6 molecules-27-01565-f006:**
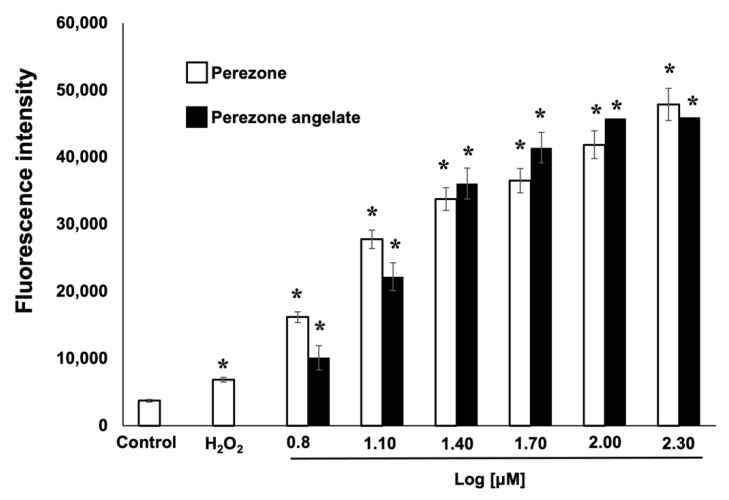
ROS production in U373 cells treated for 48 h with increasing perezone and perezone angelate concentrations evidenced by flow cytometry analysis of DCFH-DA staining (*n* = 3). Bars represent mean ± SEM. Statistical differences were calculated by one-way ANOVA, and Tukey’s post hoc test, * *p* < 0.05 (between control and treatments).

**Figure 7 molecules-27-01565-f007:**
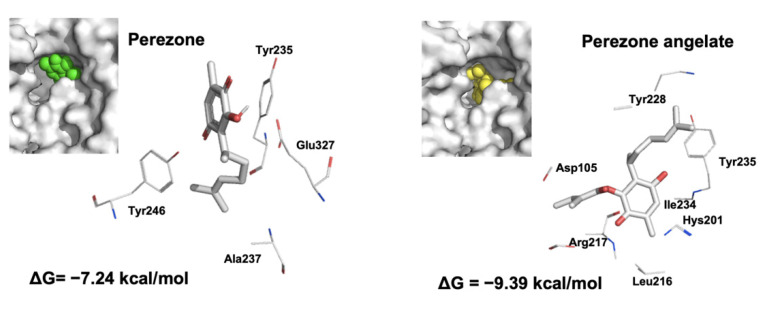
Non-bonding interactions established by perezone and perezone angelate with PARP-1, demonstrated by docking studies, the volume occupied by each compound in the catalytic site of PARP-1 is shown at the left of each image. Perezone angelate showed the highest affinity; the presence of its angelic substituent allowed it to present a different binding mode and exhibit higher affinity than perezone.

**Figure 8 molecules-27-01565-f008:**
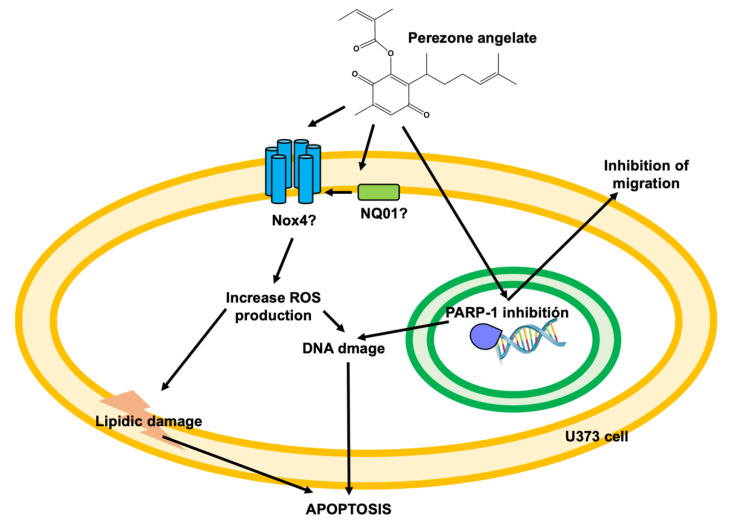
Proposed effects exerted by perezone angelate in U373 cells. Perezone angelate, similar to other quinones, can increase ROS production by Nox4 and NQO1. ROS production can produce both DNA and lipidic damage, resulting in apoptosis induction. In addition, PARP-1 inhibition avoids DNA reparation and contributes to DNA damage, thus contributing to apoptosis induction. Finally, PARP-1 inhibition can impair cell migration.

**Table 1 molecules-27-01565-t001:** Parameters joined with Lipinski’s rule of five, the theoretical values of the topological polar surface area (TPSA), and water solubility (S_w_) for perezone and perezone angelate.

Compound	MWt	Log P	Log D	HBD	HBA	TPSA	S_w_
	Expected Values						
	(≤450 g/mol)	(≤5)	(≤4)	(≤3)	(≤7)	(140 Å^2^)	(≥0.010 mg/mL)
Perezone	248.32	3.336	0.883	1	3	54.37	0.118
Perezone angelate	330.43	4.478	4.478	0	4	60.44	0.010

MWt: molecular weight; Log P: calculated logarithm of the 1-octanol-water partition coefficient (neutral species); Log D: the distribution coefficient, D, is the appropriate descriptor for non-ionized and ionizable compounds at a given pH in 1-octanol-water; HBD: hydrogen bond donor atoms; HBA: hydrogen bond acceptor atoms.

**Table 2 molecules-27-01565-t002:** Theoretical values of distribution V_d_, effective permeability P_eff_, apparent permeability MDCK, percentage of the unbound drug to blood plasma proteins %Unbnd, blood-to-plasma concentration ratio RBP, BBB filter, BBB partition coefficient logBB, and the affinity of P-glycoprotein (P-gp) for compounds perezone and perezone angelate.

Compound	V_d_	P_eff_	MDCK	%Unbnd	RBP	BBB Filter	logBB ^a^	P-gp Substrate/ Inhibitor
	Expected Values							
	(≤3.7 L/kg)	(≥0.5 cm/s·10^4^)	(≥30 cm/s·10^7^)	(>10%)	(<1)	(High/ Low)		
Perezone	0.188	6.445	399.055	7.67	0.601	High	−1.057	No/No
Perezone angelate	1.560	4.547	1106.031	11.45	0.787	High	0.459	No/Yes

^a^ According to the classification made by Ma et al. [23], high absorption to CNS: logBB more than 0.3; middle absorption to CNS: logBB 0.3–(−1.0); low absorption to CNS: logBB less than −1.0.

## Data Availability

The datasets generated during and/or analyzed during the current study are available upon request to dra.hernandez.ipn@gmail.com and nicovain@yahoo.com.mx.

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
