# Peer review of "In Vitro and Computational Studies of Perezone and Perezone Angelate as Potential Anti-Glioblastoma Multiforme Agents"

_molecules, 2022, doi:10.3390/molecules27051565_

Round 1
Reviewer 1 Report
The manuscript entitled: In Vitro and Computational Studies of Perezone and Perezone Angelate as Potential Anti-Glioblastoma Multiforme Agents, presents interesting study, yet there are number of points to be addressed:
Results
-Compare the ant proliferative activity of Perezone and Perezone Angelate with a positive standard in MTT and apoptosis assays.
-Why 100 uM and 200 uM concentrations were used in the Annexin assay and not the IC50 values (as in the cell migration assay?). There should be consistency in using the concentrations/ranges.
-Figure 3:
Include the histogram of the control.
Histograms are not clear.
The colours of the histogram and the legend are confusing: Align the legend so that it is clear referring to the circle shapes.
-Figure 4b: include x and y axis.
-Figure 5 not clear, increase font.
Methods
-Mention name of the plant from which Perezone and Perezone Angelate were extracted.
-Include reference for each of the used methods, MTT: doi.org/10.3390/molecules27010090 for example, Annexin:…
-Mention make of the flowcytomer
-Mention how many times every experiment was repeated
-Which programme was used for analysis
Conclusion
-What could be future work to develop your candidate?
Author Response
PERFORMED CHANGES, ACCORDING TO THE REFEREES RECOMMENDATIONS
Reviewer #1: The manuscript: In Vitro and Computational Studies of Perezone and Perezone Angelate as Potential Anti-Glioblastoma Multiforme Agents, presents interesting study, yet there are number of points to be addressed:
Results
- Compare the ant proliferative activity of Perezone and Perezone Angelate with a positive standard in MTT and apoptosis assay.
Response: We have included suberoylanilide hydroxamic acid (50 mM) as a positive control in MTT assays and negative control in apoptosis assay.
Lee, J.S.; Kim, H.Y.; Jeong, N.Y.; Lee, S.Y.; Yoon, Y.G.; Choi, Y.H.; Yan, C.; Chu, I.S.; Koh, H.; Park, H.T.; Yoo, Y.H. Expression of αB-crystallin overrides the anti-apoptotic activity of XIAP. Neuro Oncol 2012, 14, 1332-1345.
- Why 100 uM and 200 uM concentrations were used in the Annexin assay and not the IC50 values (as in the cell migration assay?). There should be consistency in using the concentrations/ranges.
Response: Thank you for your suggestion. We decide to include the results of the apoptosis assay employing 6.25 mM and 50 mM to perezone angelate and perezone respectively, according to its IC50 in U373 cells (6.44 mM and 51.20 mM, respectively). Results are displayed in Figure 3.
- Figure 3: Include the histogram of the control. Histograms are not clear. The colours of the histogram and the legend are confusing: Align the legend so that it is clear referring to the circle shapes.
Response: According to your suggestion, Figure 3 was reorganized. Histogram of the control was included. Additionally, the histograms displayed are bigger than those shown in the previous version-manuscript; the percentage, is included in each quadrant to avoid confusion.
- Figure 4b: include x and y axis.
Response: Labels of X and Y axis of Figure 4b were included as the reviewer suggested.
- Figure 5 not clear, increase font.
Response: Fonts of Figure 5 were increased to improve its visualization.
Methods
- Mention name of the plant from which Perezone and Perezone Angelate were extracted.
Response: The name of the plant from studied compounds were extracted was included in the methodology section as it is described below:
“Extraction of perezone and perezone angelate from the roots of Acourtia cordata was performed as previously described [13]. Physical and spectroscopic data were correlated with the literature [14]:
- Include reference for each of the used methods, MTT: doi.org/10.3390/molecules27010090 for example, Annexin:
Response: Missing data, such as references for each methodology, have been included in the present version of the manuscript based on your valuable comments.
- Mention make of the flowcytomer
Response: Missing data in the methodology section, such as the make of the flow cytomer, has been included based on your valuable comments.
- Mention how many times every experiment was repeated
Response: Missing data in the methodology section, such as number of repetitions for each experiment, has been included based on your valuable comments.
- Which programme was used for analysis
Response: Missing data in the methodology section, such as the number of repetitions for each experiment, has been included based on your valuable comments.
Conclusion
- What could be future work to develop your candidate?
Response: The proposes to continue with the anti-GBM activity of perezone angelate were included in the conclusion section as follows:
“Cell proliferation demonstrated that treatment with perezone angelate significantly induces apoptosis in U373 cells, with low cytotoxicity in rat glial cells, being apoptosis the type of cell death. The inhibition of PARP-1 and the induction of an oxidative stress state in U373 cells by perezone angelate were verified.
The performed Docking studies showed that perezone angelate established numerous interactions with the catalytic domain of PARP-1, being promoted by the orientation of angelic moiety and the side chain. ADMET studies revealed that perezone angelate exhibited a reasonable probability of being well absorbed and a high probability to permeate cells and the BBB.
Results obtained reveal the anti-GBM activity of perezone angelate, highlighting the importance of the consequent experimental evaluation of this compound by pharmaco-kinetics assays to ensure that perezone angelate cross the BBB, pharmacological evaluation, in an in vivo model of GBM, in addition to toxicological evaluation to determinate its adverse side effects.”

Reviewer 2 Report
- Kindly provide Fig 3. in high resolution as the readers are not able to see the x and y-axis. Also, mention the percentages of early and late apoptotic cells in the quadrants.
- In Figures 4 and 6, I did not see any difference in anti-cell migratory activity and ROS induction of perezone and perezone angelate. Can you explain any reason behind this?
- The authors need to provide information on the type of apoptotic pathway that perezone angelate is inducing. For analysis, authors should perform a test to study the percentage changes in mitochondrial membrane potential.
- In section 4.2, the authors should mention the solubility of the tested compounds and give in the description the details of the type of organic solvent used during the test.
- In section 4.3, the authors should mention the software name that was used to analyze the flow cytometry data.
- The author should mention the tested concentration of dyes used for MTT, apoptosis, and ROS assays.
- Authors should provide the instrument name, company, and country detail throughout the manuscript wherever, the instrument is mentioned.
- In statistical analysis, the authors should also mention the p values for other **, ***, ****.
Author Response
PERFORMED CHANGES, ACCORDING TO THE REFEREES RECOMENTATIONS
Reviewer #2:
- Kindly provide Fig 3. in high resolution as the readers are not able to see the x and y-axis. Also, mention the percentages of early and late apoptotic cells in the quadrants.
Response: According to your suggestion, Figure 3 was reorganized. Histogram of the control was included. Additionally, the histograms displayed are bigger in comparison with the previous version-manuscript, the percentage is included in each quadrant to avoid confusions.
- In Figures 4 and 6, I did not see any difference in anti-cell migratory activity and ROS induction of perezone and perezone angelate. Can you explain any reason behind this?
Response: A wide explanation of the correlation between mechanism of action of perezone angelate (ROS production and PARP-1 inhibition) and their anti-GBM effects were widely described in the discussion-section as follows:
“As reported previously, perezone exerts pro-apoptotic effects by at least two mechanisms of action: the induction of oxidative stress and PARP-1 inhibition [13]. Figure 8 shows the proposed effects of perezone angelate to explain its pro-apoptotic effects in GBM cells. Perezone and perezone angelate belong to quinones; due to this chemical characteristic, ROS production could be related to an increase in the activity of nicotinamide adenine dinucleotide phosphate (NADPH) oxidase (Nox4), as it has been demonstrated by other quinone derivatives [33]. Furthermore, NAD(P)H:quinone oxidoreductase (NQO1) has been related to Nox4 stimulation [34]. ROS production allows explaining in part the cytotoxic activity of perezone and perezone angelate due to cancer cells being exposed to a relatively high level of ROS compared to that for normal cells, primarily due to their active metabolism driven by oncogenic signals [35]. Pharmacological elevation of intracellular ROS represents an effective strategy to target cancer cells selectively. An exogenous source of ROS insult that is within a tolerable level to normal cells could exceed the threshold that cancer cells can endure, leading to selective eradication of cancer cells [36]. Then, excessive ROS levels irreversibly could damage DNA and lipids and ultimately produce apoptosis of cancer cells [37, 38].
Figure 8. Proposed effects exerted by perezone angelate in U373 cells. Perezone angelate, similar to other quinones, can increase ROS production by Nox4 and NQO1. ROS production can produce both DNA and lipidic damage resulting in apoptosis induction. In addition, PARP-1 inhibition impairs DNA reparation and contributes to DNA damage, thus contributing to apoptosis induction. Finally, PARP-1 inhibition can impair cell migration.
In addition, it has been shown that perezone produced a reduction in the mitochondria membrane potential in a dose-dependent manner that correlates with its cytotoxic activity. In this sense, perezone and its derivatives could alter the mitochondrial electron transport, inducing apoptosis intrinsic pathway [39].
Although ROS production its very similar by perezone and perezone angelate, the highest inhibition of PARP-1 by perezone angelate explains its high cytotoxic activity. Particularly, cancer cells have defects in DNA repair pathways, then tumor cells are susceptible to PARP-1 inhibition [40]. Finally, PARP-1 inhibition could influence cell migration, as previously reported [41].”
- The authors need to provide information on the type of apoptotic pathway that perezone angelate is inducing. For analysis, authors should perform a test to study the percentage changes in mitochondrial membrane potential.
Response: According to the previous suggestion, we decide to explain the pathways that favor apoptosis induction by perezone angelate in U373 cells in the discussion section. Additionally, we included the description of the previous manuscript that demonstrate changes in mitochondrial membrane potential induced by perezone.
“In addition, it has been shown that perezone produced a reduction in the mitochondria membrane potential in a dose-dependent manner that correlates with its cytotoxic activity. In this sense, perezone could alter the mitochondrial electron transport, inducing apoptosis intrinsic pathway [39].”
- In section 4.2, the authors should mention the solubility of the tested compounds and give in the description the details of the type of organic solvent used during the test.
Response: Missing data in the methodology section, such as organic solvent using during the test was included based on your valuable comments:
“100 mL of solutions with increasing concentrations of perezone and perezone angelate diluted in fresh medium containing ethanol 2% were added to the respective wells, obtaining a final concentration of 6.25, 12.5, 25, 50, 100, 200 and, 400 mM (n=8) and incubated for an additional 48 h (final concentration of ethanol 1%).”
- In section 4.3, the authors should mention the software name that was used to analyze the flow cytometry data.
Response: Missing data in the methodology section, such as software to analyze flow cytometry results, has been included based on your valuable comments.
- The author should mention the tested concentration of dyes used for MTT, apoptosis, and ROS assays.
Response: Missing data in the methodology section, such as concentrations of dyes employed to perform MTT, apoptosis and ROS assays, have been included based on your valuable comments.
- Authors should provide the instrument name, company, and country detail throughout the manuscript wherever, the instrument is mentioned.
Response: Missing data in the methodology section, such as information for each instrument employed, has been included based on your valuable comments.
- In statistical analysis, the authors should also mention the p values for other **, ***, ****.
RESPONSE: Missing data such as p values in Figure 3 for each instrument employed, has been included based on your valuable comments.

Round 2
Reviewer 1 Report
Thank you for making amendments
Reviewer 2 Report
The manuscript is revised very well and the authors have done all the necessary changes. Now, this manuscript can be published.